# Ground-Based Hyperspectral Remote Sensing for Estimating Water Stress in Tomato Growth in Sandy Loam and Silty Loam Soils

**DOI:** 10.3390/s21175705

**Published:** 2021-08-24

**Authors:** Kelvin Edom Alordzinu, Jiuhao Li, Yubin Lan, Sadick Amoakohene Appiah, Alaa AL Aasmi, Hao Wang, Juan Liao, Livingstone Kobina Sam-Amoah, Songyang Qiao

**Affiliations:** 1College of Water Conservancy and Civil Engineering, South China Agriculture University, Guangzhou 510070, China; kelvinedomalordzinu@gmail.com (K.E.A.); aasadick07@gmail.com (S.A.A.); alaaasmi83@gmail.com (A.A.A.); whao20000904@gmail.com (H.W.); qsy1999@stu.scau.edu.cn (S.Q.); 2National Center for International Collaboration Research on Precision Agricultural Aviation Pesticides Spraying Technology (NPAAC), College of Electronic Engineering, South China Agriculture University, Guangzhou 510070, China; ylan@scau.edu.cn; 3College of Engineering, South China Agriculture University, Guangzhou 510070, China; liaojuan0529@scau.edu.cn; 4Department of Agricultural Engineering, School of Agriculture, College of Agriculture and Natural Sciences, University of Cape Coast, Cape Coast 03321, Ghana; lsam-amoah@ucc.edu.gh

**Keywords:** crop water-stress, tomato, sandy loam, silty soils, ASD hyperspectral reflectance, water stress indicators

## Abstract

Drought and water scarcity due to global warming, climate change, and social development have been the most death-defying threat to global agriculture production for the optimization of water and food security. Reflectance indices obtained by an Analytical Spectral Device (ASD) Spec 4 hyperspectral spectrometer from tomato growth in two soil texture types exposed to four water stress levels (70–100% FC, 60–70% FC, 50–60% FC, and 40–50% FC) was deployed to schedule irrigation and management of crops’ water stress. The treatments were replicated four times in a randomized complete block design (RCBD) in a 2 × 4 factorial experiment. Water stress treatments were monitored with Time Domain Reflectometer (TDR) every 12 h before and after irrigation to maintain soil water content at the desired (FC%). Soil electrical conductivity (Ec) was measured daily throughout the growth cycle of tomatoes in both soil types. Ec was revealing a strong correlation with water stress at *R*^2^ above 0.95 *p* < 0.001. Yield was measured at the end of the end of the growing season. The results revealed that yield had a high correlation with water stress at *R*^2^ = 0.9758 and 0.9816 *p* < 0.01 for sandy loam and silty loam soils, respectively. Leaf temperature (LT °C), relative leaf water content (RLWC), leaf chlorophyll content (LCC), Leaf area index (LAI), were measured at each growth stage at the same time spectral reflectance data were measured throughout the growth period. Spectral reflectance indices used were grouped into three: (1) greenness vegetative indices; (2) water overtone vegetation indices; (3) Photochemical Reflectance Index centered at 570 nm (PRI_570_), and normalized PRI (PRInorm). These reflectance indices were strongly correlated with all four water stress indicators and yield. The results revealed that NDVI, RDVI, WI, NDWI, NDWI_1640_, PRI_570_, and PRInorm were the most sensitive indices for estimating crop water stress at each growth stage in both sandy loam and silty loam soils at *R*^2^ above 0.35. This study recounts the depth of 858 to 1640 nm band absorption to water stress estimation, comparing it to other band depths to give an insight into the usefulness of ground-based hyperspectral reflectance indices for assessing crop water stress at different growth stages in different soil types.

## 1. Introduction

Plant response to water stress is articulated by a variety of physiological and biophysical changes as well as soil characteristics (chemical and physical properties). Due to climate change and increasing global water scarcity, water stress has been the most perilous abiotic stressor to plant growth. However, to increase food production using a minimal amount of water, timely detection and quantification of crop water status in crop production are very vital in precision agriculture technology [1,2,3].

Remote sensing, as a tool in precision agriculture to effectively schedule irrigation in vegetable production, has not been extensively developed, especially in greenhouse vegetable production. For sustainability of agricultural water management, timely identification of plant abiotic stress is required [4], to improve food and water security. It is possible to relate available soil water to crop leaf water content if the density and water content of the crop leaf is highly correlated to the available soil water [5,6,7,8].

A remote sensing approach in assessing crop water status is fast, accurate, less laborious, and non-destructive at different scales (ground, airborne, and satellite) of measurements [4,9,10,11]. According to Ihuoma and Madramootoo [4], Streher et al. [12], Zhang et al. [13] and Nemeskéri et al. [14], hyperspectral spectra reflectance extending from the visible to the near-infrared to the midway infrared can provide spectral features concerning differences in the physiology, biophysical, and biochemical composition of crop leaf. Zhang and Zhou [15] and Zhou et al. [16], reported that hyperspectral reflectance in assessing crop water stress is very important for crop yield forecast. The technique for remote sensing of crop water stress has included crop leaf temperature and other vegetation index determination using red and near-infrared reflectance. Remote sensing of crop leaf is mainly performed by obtaining the electromagnetic wave reflectance information from leaves using passive sensors [9,17,18,19,20]. According to Katsoulas et al. [8] and Honkavaara et al. [21], the reflectance of light spectra from plants varies with plant type, the growth stage of the crop, water content within tissues, and other inherent factors. Moreover, spectral reflectance characteristics of crop leaves are mostly dependent on the biochemical, biophysical, and morphological characteristics of the leaves [5,22,23].

Hyperspectral remote sensing of crop leaf using an Analytical Spectrometry Device (ASD) is based on the following spectra band regions: (i) the ultraviolet region (UV), which ranges from 10 to 380 nm; the visible spectra region, which ranges from 380 to 740 nm; the near-infrared band (740–1400 nm), and 1400 to 2500 nm ranges for the short wavelength region [24,25,26]. According to Hatfield et al. [27], the surface of leaves’ absorptivity or emissivity in the thermal waveband of a healthy and fully grown green plant is generally between the range of 0.96–0.99, whereas dry or unhealthy plant emissivity rate generally ranges from 0.88 to 0.94. The use of hyperspectral reflectance to assess plant water stress or physiological changes is very complicated, as the mechanisms linking spectra reflectance and emission to plant functional traits are not always clear or known [26]. Spectroscopy techniques coupled with deep learning algorithms have been used for leaf morphological and biochemical traits in crop photosynthesis potential estimation, leaf chlorophyll content estimation, sensitivity in PRI water stress assessment, NDWI, WI, SRWI vegetation indices for water stress assessment as well as structure vegetation indices such as NDVI, RDVI, mNDVI_750_, and many other VIs (Ihuoma and Madramootoo [4], Ustin et al. [5], Gao [6], Gamon et al. [10], Zhang et al. [13], Nemeskéri et al. [14], Zarco-Tejada and Ustin [17], Roujean and Breon [28], Gitelson et al. [29], Sakamoto et al. [30], Peñuelas et al. [31], Chen et al. [32], Chen et al. [33], Liu et al. [34], Mzid et al. [35], Rouse et al. [36], Zeng et al. [37]), to evaluate water stress based on leaf reflectance successfully by using specific wavelengths or indices related to crop water status of crops at different growth stage and water treatment levels. Vegetation indices that measure crop canopy greenness are sensitive to both leaf area and chlorophyll content [3,5].

However, the pigment-based indices such as NDVI, RDVI, CL_green_ help researchers to better understand the biophysical and biochemical processes of crop leaves and also help in predicting crop yield (Katsoulas et al. [8], Zhang et al. [9]). According to Zhang et al. [38], water index vegetation indices such as WI, NDWI, NDWI_1640_, NDWI_2130_, and SRWI are very essential in the estimation of leaf temperature, leaf relative water content, leaf area index (LAI), yield, and end of season biomass within 980 and 1240 nm. The photochemical index is a very good indicator of photosynthetic rate and spatiotemporal scales, this has been used by (Fischlin et al. [39], Magney et al. [40], Garbulsky et al. [41], Wong and Gamon [42], Zhang et al. [43]), for the detection of the reactivation of photosynthesis in stress evergreen species.

The effects of seasonal drought on the photosynthetic apparatus have also been detected by satellite-based PRI. Maximum CO_2_ assimilation has also been efficiently estimated by PRI under severe drought conditions and this has established that changes in PRI were correlated with water stress in maize (Zhang et al. [9]). Photosynthetic variability induced by heat and drought is simultaneously accompanied by complex physiological and biochemical processes, which could constrain the PRI-based estimation of the photosynthetic apparatus (Zhou et al. [16], Magney et al. [40], Garbulsky et al. [41], Wong and Gamon [42], Zhang et al. [43], Gamon et al. [44], Massonnet et al. [45] and Pedrol et al. [46]). For optimization of tomato productivity, adequate and well-planned irrigation scheduling is required to ensure adequate water supply during the entire growth cycle. It is very necessary to make very good irrigation scheduling management decisions to reduce the impact of global warming, climate change, drought, and the high competition of other water users on the high demand of the limited availability of freshwater resources for irrigation. According to Afzal et al. [47], precise irrigation scheduling is essential to optimize irrigation water use, improve crop yield, and avoid excessive irrigation that may result in yield, water loss [4], or leaching of agricultural nutrients that would degrade soil and water. However, implementation of this novel strategy requires accurate information on plant water status [38,48,49]. This study focuses on estimating water stress on tomato growth in different soils at different field capacity (FC%) using greenness vegetation indices, water indices, and photochemical indices and comparing it with crop water stress indicators (LT °C, RLWC, LAI, LCC) and yield. Results from this study have proven a strong correlation between the water stress indicators and vegetation indices (VI’s) obtained from ASD hyperspectral reflectance for estimating water stress in tomatoes grown in different soil texture types and has clearly explained and offered hypothetical management decisions for optimizing water productivity and rapid identification of water stress in greenhouse tomato cultivation.

## 2. Materials and Methods

### 2.1. Study Area and Experimental Design

Pot experiments were conducted in a greenhouse at Tea Research Academy, South China Agricultural University, Guangzhou—China from October 2020 to March 2021. The research site is located between 23°157′826″ N and 113°350′668″ E, with an elevation of 30 m. A simple arch double-span greenhouse was used with a gutter length of 26 m, a width of 20 m, a gutter height of 3 m, gable height of 4.5 m. The top was covered with diffused polyethylene film with 95% light transmission, the sides covered with 50 mesh insert net with the floor covered with black woven polyethylene mulch. The experimental plot was a 5 m × 8 m field and the experimental design was a 2 × 4 factorial experimental design arranged in a randomized complete block design (RCBD) with four replications in each sub-block. Water stress treatments applied ranged from 70–100, 60–70, 50–60, and 40–50% field capacity with each replicated four times to give a total of 16 tomato plants in each sub-block (32 plants for the main block). Fertilizer was applied based on the available N:P:K observed from the soil analysis in each soil type to meet crop nutrient requirement. Soil physical and chemical properties are shown in Table 1. Tomato (*Solanum Lycopersicum*) cv. Xiang Sheng seed was nursed in a 6 × 12 nursery tray in cocopeat on 15 October 2020 and transplanted on 10 November 2020 into truncated black plastic pots (single plant per pot) with 1 m × 0.4 m plant spacing within a block and 1.5 m between block spacing; black plastic mulch was used to cover the ground to control weeds.

### 2.2. Water Stress Treatment

Irrigation water was applied through a gravity-driven drip system, with a discharge of 1.2 L/h, and the flow rates were calibrated in the field. Water stress treatment was applied based on the soil field capacity. Soil moisture content (SMC) in each pot was continuously measured with soil moisture sensors Time Domain Reflectometers (TDR). The water stress treatment levels for each treatment used were 70–100% FC, 60–70% FC, 50–60% FC, and 40–50% FC. The upper irrigation and lower irrigation thresholds were set at 10% depletion of soil FC% for each treatment. The volumetric soil water content and the observed soil moisture content measured by a Time Domain Reflectometry (IMKO. TRIME. PICO TDR HD2 64) every 12 h before and after irrigating the plants was used to determine the soil moisture content before water application. The probes of TDR were inserted throughout the soil medium in the bucket at a depth of 30 cm. The volume of water applied to soil to bring the soil back to field capacity was calculated based on the following equation by Hamouda et al. (2019),
(1)$$I=\frac{Q+\left(SWC-AWC\right)\times DSM}{A}\times 1000,$$
where *I* is the irrigation water (mm), *Q* is the volume of ponding water (mm), *SWC* is the saturated water content of the soil (%), *AWC* is the actual water of the soil when irrigating (%), *DSM* is the dry soil mass (kg), and *A* which is the area of a truncated cone is given by the equation
(2)$$A=\pi \left(R^{2}+r^{2}\right)+\pi \left(R+r\right)\sqrt{h^{2}+{\left(R-r\right)}^{2}},$$
where: *R* is the radius of the bottom base; *r* is the radius of the top base and *h* is the slant height of the truncated cone. High-quality irrigation water was applied through a drip system, with emitters placed in each pot. Water stress treatment was introduced 2 weeks after transplanting based on soil % FC.

### 2.3. Soil Water Content Measurement

Soil water content was measured using the TDR and the Gravimetric technique. These two techniques measured different values of soil water content for the same soil type at the same field capacity. A correlation was used to establish a relationship between the TDR readings and the gravimetric reading to give a good reflection of Available Soil Water Content (AWC) for the two soils used for the experiment to maintain continuous and constant FC% for each treatment [50]. The comparison of TDR probe moisture values and gravimetric moisture values plotted against each other revealed a very high correlation between TDR probe moisture values and gravimetric moisture values estimated in the laboratory.

### 2.4. Plant Nutrition and Pest and Diseases Management

Fertilizer applications were performed twice during the planting season (on the 20 November onset of the vegetative stage) with 20-20-20 N-P-K water-soluble fertilizer, at a rate of 3.5 kg of N per hectare for both soils due to the high levels of NPK already available in the soil. This was changed to calcium nitrate after the first fruits were noticed and later changed to potassium nitrate during the fruits’ expansion stage. Organic pesticide (neem oil plus emulsifying soap) was applied in a ratio of 2:1 in 15 L of water biweekly. Weeds were handpicked from pots and woven polyethylene plastic mulch was used to control weeds on the greenhouse floor.

### 2.5. Field Measurements

Plant biotic stress was based on the physiological parameters of the plant and this was measured at each growth stage of the growth cycle (vegetative stage, anthesis stage, fruit development stage, and senescence stage) for the two soil types that were used in the experiment. Parameters measured include leaf temperature (LT °C), leaf chlorophyll content (LCC), leaf area index (LAI), and leaf relative water content (LRWC). Crop leaf temperature was measured with a Raytek^®^ handheld non-contact infrared thermometer with an emissivity of 0.95 Wm^−2^ and a display resolution of 0.2 °C (0.5 °F). The thermometer was held 30 cm above the tomato leaf with the laser point set at an angle of 90° to the horizontal [4,51]. Infrared thermometer measurements were taken at each growth stage of the tomato growth cycle on five sampled plants in each treatment block. Leaf temperatures were measured at four viewing directions (north, south, east, and west) and average temperatures for each treatment were calculated. Temperature readings were taken between 11:00 a.m. and 2:00 p.m. to ensure maximum sunlight intensity in the greenhouse and on the plants’ leaves as recommended by Ihuoma and Madramootoo [4]. Leaf relative water content (LRWC) was measured at each stage of plant growth by sampling young and fully expanded leaves from 4 plants in each treatment. The leaf samples were enclosed in a plastic zip bag and kept in a cooling chamber at 5 °C and transported to the laboratory. Fresh weight (FW) was recorded using an electronic scale and the weight was recorded. Samples were immersed in distilled water for 72 h, blotted, and weighed to obtain the turgid weight (TW). Finally, leaf samples were dried at a temperature of 72 °C in an oven, until constant dry weight (DW) was achieved. Leaf relative water content was calculated using the equation by [4,18,52],
(3)$$LRWC=\frac{\left(FM-DM\right)}{\left(TM-DM\right)}\times 100,$$
where, *LRWC*% = Leaf Relative Water Content, *FM* = fresh leaf mas, *DM* = dry leaf mass, *TM* = turgid leaf mass

Leaf chlorophyll content was measured using a Handheld Konica Minolta SPAD 502 chlorophyll meter. SPAD measurements were taken on the newest fully expanded leaf at each growth stage of the tomato growth stage. Reading was taken on a designated leaf on all plants approximately half the distance of each leaf and halfway between the leaf edge [29,53]. SPAD data was converted to N-sufficiency Index using a plant with no water stress as the reference plant using the equation by [54]
(4)$$NSI=\frac{SPAD_{target}}{SPAD_{reference}},$$

Leaf Area Index (LAI) was determined using the destructive method, all leaves were removed separately from sampled plants (4 plants) from each treatment at each growth stage of tomato growth, and 12 to 20 leaves were collected per plant. Leaves were placed into a rectangular sketch of white paper. Leaf area was calculated using grid or graph paper technique, a linear regression model of leaf area (*LA*) developed [23]. The leaf area of the tomato was related to variable leaf dimensions (*LL* and *LW*) using an equation by [23]
(5)$$LA=0.00409\times \left(192.68\times \left(\frac{LL}{100}\right)\times \left(\frac{LW}{100}\right)-1\right),$$
where, *LA*, Leaf area (m^2^), *LL*, Leaf length (cm), and *LW*. Leaf Area Index (*LAI*) using the equation by [55].
(6)$$LAI=\frac{\mathrm{LAm}\times N}{A},$$
where *N* is the number of leaves on the plant and *A* the area (cm^2^) occupied by one plant in the cropped area.

### 2.6. Hyperspectral Data Acquisition and Processing

Hyperspectral remote sensing is the measurement of reflected radiance on the narrow continuous spectral bands over the full visible and solar reflective spectrum. Tomato leave reflectance was measured using Analytical Spectral Device (ASD) Field Spec 4 (FS4), a non-image spectrometer. The spectral range of this highly resolving device covers wavelengths between 350 and 2500 nm, spectral resolution varying from 3 to 700 nm in the very short and 10 to 1400/2100 nm in the further wavelengths, with a sampling interval of 1.4 nm at 350–1050 nm and 2 nm at 1000–2500 nm, which has a scanning time of 100 milliseconds. This hyperspectral device (ASD) has stray light specification of VNIR 0.02%, SWIR 1 and 2 is 0.01%, wavelength reproducibility 0.1 nm, wavelength accuracy 0.5 nm, maximum radiance VNIR 2× Solar, SWIR 10× Solar. The device records spectra based on the information of 2151 bands. The ASD has three detectors which include 1. VNIR detector (350–1000 nm):512 element silicon array; 2. SWIR 1 detector (1000–1800 nm): Graded Index InGaAs Photodiode, TE Cooled; 3. SWIR 1 detector (1800–2500 nm): Graded Index InGaAs Photodiode, TE Cooled. Each detector has noise equivalent radiance (NEdL) of 1.0 × 10^−9^ W/cm^2^ /nm/sr at 700 nm, 1.2 × 10^−9^ W/cm^2^ /nm/sr at 1400nm, and 1.9 × 10^−9^ W/cm^2^ /nm/sr at 2100 nm, respectively. The ASD with an input of 1.5 m fiber-optic (25° field of view) with an optional narrow field of view fiber optics was calibrated for tomato leaf reflectance measurement using the reflectance from a whiteboard (Spectralon panel). The pistol was focused perpendicular to the plant leaf at a target distance of 100 cm to achieve a ground area of 44 cm in diameter. This was performed following the recommendation of Danner et al. (2015) (24). The instrument was controlled on an IBM Lenovo intel^®^ Core™ 2 Duo CPU T7100@1.80 GHz, 1.79 GHz, 0.97 GB of RAM. With windows^®^ 7 64-bit. ViewSpec Pro was used to process reflectance data gathered by the RS³ software.

#### Spectral Vegetative Indices

Narrow-band spectral vegetation indices (SVI’s) which have revealed the possibility for evaluating the characteristics of vegetation parameters related to plant physiology, morphology, and biochemistry were preselected for analysis. These spectral transformations of two or more bands designed to enhance the contribution of vegetation properties have reliable spatial and temporal intercomparisons of crops’ photosynthetic activity and canopy structural variations. It is a very powerful tool in assessing changing climate or global warming. The VIs used in this study were grouped into three, which include (i) greenness vegetation indices, (ii) water content vegetation indices, and (iii) photochemical vegetation indices. Four greenness vegetation indices (NDVI, mNDVI_750_, RDVI, CL_green_; Five (5) water content vegetation indices WI, WBI, SRWI, NDWI, NDWI_1640_, NDWI_2130_, and two (2) photochemical reflectance vegetation indices, PRI and PRInorm, were used in this study. These are represented in Table 2.

### 2.7. Statistical Analysis

One-way analysis of variance was used to analyze differences between crop water stress indicators (leaf temperature (LT °C), relative leaf water content (RLWC) %, leaf chlorophyll content (LCC) mg g^−1^, and leaf area index (LAI) m^2^/m^2^) of the tomato plant and spectral vegetative indices (VIs) (NDVI, mNDVI_750_, RDVI, CL_GREEN_, WI, WBI, SRWI, NDWI, NDWI_1640_, NDWI_2130_, and PRI among treatments and blocks. IBM SPSS Statistics 21 package was used for the statistical analysis. Pearson correlation coefficient in a two-tailed test of significance was used to describe the effect of soil field capacities of the two soil types on tomato because they were continuous scale variables that were also normally distributed. The correlation assumes a value of (−1 and +1) which explains the variance in data concerning water stress. A negative value means perfect negative correlation, a positive value means perfect positive correlation. One-way analysis of variance (ANOVA) was conducted and Fisher’s Least Significant Difference was used to separate the significant differences among treatments at 0.001, 0.01, and 0.05 significant levels. Linear regression analysis was used to determine the relationship between spectral reflectance variables and water stress indicators in a bivariate analysis in a two-tailed test run in IBM SPSS statistic 21.

## 3. Results

### 3.1. Soil Water Content Measurement

Results from this study revealed that TDR and gravimetric soil water content measurement had a strong correlation. The correlation coefficient turned out to be *R*^2^ = 0.9685 and 0.9212, *p* < 0.05 for sandy loam and silty loam soils, respectively, as shown in Figure 1. The irrigation water applied to tomatoes throughout the growing cycle for each growth stage for the different treatments in sandy loam soil and silty loam soil is shown in Table 3.

Variation in soil water and amount of irrigation water applied during the growing season of tomato growth in sandy loam and silty loam soil indicates a higher irrigation water application during the anthesis stage and fruits development and expansion stages of the plants’ growth. This is the stage in the crop’s growing cycle where biophysical, biochemical, and physiological processes are critical and requires more water to optimize crop productivity. Daily volumetric soil water content for various water stress treatments for both soils is shown in Figure 2a,b.

### 3.2. Effect of Water Stress on Electrical Conductivity (Ec) and Marketable Yield

The effects of water stress treatments on *Ec* and marketable yield of tomato are shown in Figure 3. The total marketable yield ranged from 6.8 to 2.02 kg/plant, for 70–100% FC and 40–50% FC treatment, respectively, in sandy loam soil and 5.4–1.75 kg/plant for 70–100% FC and 40–50% FC treatments in silty loam soil. The highest mean marketable yield was obtained from 70–100% FC treatment (6.8 kg/plant) in sandy loam soil while the least marketable yield was obtained from 40–50% FC treatment (1.78 kg/plant) in silty loam soil. The average yield from the 70–100% FC treatment in sandy loam soil was not statistically different from the yield obtained from 60–70% FC but significantly different from the yield obtained from 50–60% FC and 40–50% FC treatments at *p* < 0.05. However, the average yield obtained from the 70–100% FC treatment in silty loam soil was statistically different from the average yield obtained from 60–70% FC, 50–60% FC, and 40–50% FC treatments at *p* < 0.01 *p* < 0.01 and *p* < 0.05, respectively. Results indicated that sandy loam and silty loam have different field capacities, yield from sandy loam soil was significantly different from silty loam soils at *p* < 0.001. This implies that all soils have their own FC % threshold, below which if stress occurs it causes an evident decrease in crop yield. However, sandy loam, silty loam, and soil electrical conductivity showed a strong correlation with water stress treatment above 0.95 at *p* < 0.001.

### 3.3. Crop Spectral Signatures

Crop spectral signatures obtained from tomato leaves with ASD hyperspectral spectrometer from water stress treatments in silty loam and sandy loam soils are shown in Figure 4a,b, respectively. The plant canopy reflectance varied among water stress treatment and soil type. The spectral signatures displayed similar trends at each growth stage and throughout the entire tomato growth cycle. The spectral signature followed a similar pattern for each of the water stress treatments for both silty loam and sandy loam soils. Soil% FC at 70–100% FC recorded the highest reflectance value whilst 40–50% FC recorded the lowest reflectance values in the NIR spectral range (740–1400 nm), respectively. The spectral reflectance curves show peaks near 1015 and 1135 nm, and trough near 1077 nm. High water-stressed crops exhibited lower reflectance values within the NIR spectral range.

### 3.4. Vegetation Indices versus Water Stress Indicators and Yield

Figure 5 and Table 4 show the coefficient of determination (*R*^2^) of the linear relationships between, LT °C, LAI, LRWC (%), leaf chlorophyll content (LCC) (mg g^−1^), and vegetation indices calculated from the ASD spectrometer in sandy loam and silty loam soils. The results show that NDVI, RDVI, CL_green_, WI, NDWI, NDWI_1640_, NDWI_2130_ PRI_570_, and PRI_norm_, were significantly correlated with yield in both sandy loam and silty loam soil (all *R*^2^ were greater than 0.45). However, NDVI showed improved performance with the most significant correlation with all four water stress indicators in both sandy loam soil at (*R*^2^ = 0.75, *p* < 0.001 for LT °C; 0.81, *p* < 0.001 for LAI; 0.89, *p* < 0.001 for LRWC; 0.89, *p* < 0.001 for LCC) and silty loam soils at (*R*^2^ = 0.79, *p* < 0.001 for TL °C; 0.70, *p* < 0.001 for LAI; 0.67 *p* < 0.001 for LRWC; and 0.87, *p* < 0.001;) for water stress detection. Similarly, NDVI, RDVI, CL_green_, NDWI, NDWI_1640_, and PRI_570_ had a strong positive correlation with leaf chlorophyll content (LCC) at (R^2^ = 0.80, *p* < 0.001; 0.71, *p* < 0.001; 0.62, *p* < 0.00; 0.83, *p* < 0.001; 0.75, *p* < 0.001 and 0.58 *p* < 0.001) and (*R*^2^ = 0.87; 0.60; 0.52; 0.87; 0.70, *p* < 0.001; and 0.62 all at *p* < 0.001) for sandy loam and silty loam soils, respectively. Similarly, WI, NDWI, NDWI_1640,_ PRI_570,_ PRI_norm,_ were strongly correlated with RLWC at (*R*^2^ = 0.64, 0.72, 0.61; 0.73; 0.74. and 0.67; 0.75; 0.54; 0.69; 0.70 *p* < 0.001) for sandy loam and silty loams, respectively. While NDVI, RDVI and CL_green_ were best correlated with yield, leaf chlorophyll content (LCC), and leaf area index (LAI), water content vegetation index and photochemical index were best correlated with RWC and LT °C. 

However, the PRI_norm_ and PRI _570_ revealed an improved correlation with all the measured water stress indicators and yield at *R*^2^ value above 0.35 for both sandy loam and silty loam soils at significance levels.

## 4. Discussion

### 4.1. Soil Water Content Measurement

The soil water content measured in this study revealed a strong positive correlation between TDR moisture values measured in the field and gravimetric moisture values estimated in the laboratory for sandy loam and silty loam soils. The correlation coefficient turned out to be *R*^2^ = 0.9628 for sandy loam soil and *R*^2^ = 0.9212 for silty loam soils. As a result of the statistical analysis, the relationship between the gravimetric method and the TDR measurements was found to be significant (*p* < 0.05). The result of the analysis suggested that there was a strong relationship among the measurements. The findings of the present study are similar to the studies conducted by Abdullah et al. [50], Berni et al. [56], Chung et al. [57], Tanriverdi [58] and Degirmenci et al. [59] who reported that there was a strong relationship between the gravimetric and TDR measurement methods at R^2^ = 0.8–0.98 (*p* < 0.05). According to Liu et al. [34], Topp [60], Matema [61], Sun and Young [62], a strong relationship between the gravimetric soil moisture measurement and that of the TDR equipment yielded very good results in determining the soil water content and the difference between the measurements performed by gravimetric and TDR equipment was found to be statistically significant in terms of reducing labor and time (*p* < 0.01) [58]. Due to the strong correlation coefficient between the gravimetric water content and TDR water content value, TDR equipment can be straightforwardly used in measuring soil water content in sandy loam and silty loam soils to save a significant amount of time and labor as compared to the gravimetric measurement method, which is laborious and very destructive [50,59,63,64,65].

### 4.2. Effect of Water Stress on Ec and Yield

The changes in water content and the resultant increase in Ec between the wetting and drying periods were relatively minor. However, soil electrical conductivity (Ec) increased with reducing soil AWC. Water stress treatments applied in this study caused a reasonable to severe stress on the plants at each growth stage. High water stress resulted in high Ec levels which affected plant water uptake from the soil, hence reducing the transpiration rate of the leaves, leaf photosynthetic efficiency, and crop yields. According to Nemeskéri et al. [66], in limited water supply conditions, plants reduce the intensity of transpiration and close stomata. As a result, the canopy temperature starts to rise. Tomato crop exposed to 40–50% FC treatment recorded the lowest crop yield (yield loss of 70.3 and 67.1%) and high Ec value of 2.6 and 3.0 µs for sandy loam and silty loam, respectively, compared to the highest yield obtained from the 70–100% FC treatment for both sandy loam and silty loam soils. However, high water stress had significant effects on soil Ec (salinity) and tomato yields in this experiment as shown in Figure 3. Results from this research are similar to the results of Zhang et al. [13], Liu et al. [34], Mohamed and EL-Aziz [66], Nemeskéri et al. [67], who reported that induction of severe water stress significantly reduces the yield of tomato plants. Nemeskéri et al. [14], Chen et al. [33], Patanè et al. [68], Pék et al. [69], Helyes et al. [70], also reported that the best marketable yields of tomato are always reached with the highest water supply rate. The highest marketable yields recorded in this study were under the 70–100% FC water stress treatment (full irrigation) with a value of 170 t/ha and 135 t/ha for sandy loam and silty loam soils, respectively, which is no different from the findings of Takács et al. [71] and Lu et al. [72], who suggest that the rate of regulated deficit irrigation should be no less than 50% of water demand because, under this level, the water stress is too severe. However, water stress treatment had a strong correlation with crop yield (*R*^2^= 0.9758 and 0.9816 *p* < 0.01) for sandy loam and silty loam soils, respectively, and also had a strong correlation with Ec (*R*^2^ = 0.9651, *p* < 0.01). This was confirmed by Ihuoma and Madramootoo [4] in narrow-band reflectance indices for mapping the combined effects of water and nitrogen stress in field-grown tomato crops. The result rationalizes the need for adequate application of water concerning the soil type and FC% to enhance the productivity of tomatoes. Ihuoma and Madramootoo [4], also suggested that effects of prolonged water and nitrogen stress might have induced structural and morphological changes in the plants, such as changes in leaf structure, shape, and leaf thickness, which might be the cause of reducing yield. Gitelson et al. [29], Zhang et al. [43], Nemeskéri et al. [14,67], Patanè et al. [68], Takács et al. [71], and Chakroun et al. [73], also reported a significant correlation between water stress and plant yield, biomass, leaf area index, and chlorophyll content, which is consistent with the results of this study.

### 4.3. Vegetation Indices versus Water Stress Indicators and Yield

The imposed water stress treatments in this study were very adequate to help estimate crop water status for the advancement and authentication of spectral reflectance indices to detect water stress in tomatoes. This study revealed that NDVI and RDVI were the greenness vegetative indices that had the best correlation with water stress indicators, leaf temperature (LT °C), RLWC, LAI, and yield for sandy loam and silty loam soils. Finding by Ihuoma and Madramootoo [4] also confirmed that NDVI and RDVI revealed a good correlation with yield and chlorophyll content in estimating crop water stress. The relationship between NDVI, RDVI, and water stress indicators may be due to the dynamics of plant canopy parameters. The results from this study were also in agreement with the findings of Ustin et al. [5], Gao [6], Katsoulas et al. [8], and Sakamoto et al. [30], who reported that narrow-band and broad-band indices for assessing crop water status are very sensitive to water stress. The high correlation between NDVI, RDVI, and water stress indicator was also confirmed in the findings of Mzid et al. [35], Rouse et al. [36], Wittamperuma et al. [74] and Tittebrand and Spank [75]. NDVI and RDVI, according to Ustin et al. [5] and Zarco-Tejada and Ustin [17], are the most popular SVI for biophysical parameter retrieval, and are very sensitive in water stress and chlorophyll estimation. In addition, WI, NDWI, and NDWI_1640_ were the water index vegetation indices that had a high correlation with water stress indicators used in this study for both soil types. This high correlation can be explained by the band length (reflectance near 980 or 1240 nm) which corresponds to strong water absorption and the reflectance near 860 nm provides the datum for comparing spectra. However, WI, NDWI measure plant water content that is mostly held in the leaves that is actual liquid water in leaves instead of vapor. The results are in agreement with Ustin et al. [5], Gao [6], Katsoulas et al. [8], Zarco-Tejada and Ustin [17], Chen et al. [33] and Huete et al. [76]. PRI_norm_ was the photochemical vegetation index that showed a high correlation with all the vegetation. The correlation between PRI_norm_ and water stress indicators can be explained by the interactions among the eco-physiological indicators of plant water stress, as explained by Zhang et al. [9], Magney et al. [40], Garbulsky et al. [41], Wong and Gamon [42], and Gamon et al. [44]. However, PRI_570_ and water stress indicators had a fairly good relationship by improving on the sensitivity of detecting water stress in tomatoes. Equally, this study has shown that all vegetation indices showed a correlation with water stress indicators but NDVI, RDVI, WI, NDWI, NDW_1640,_ PRI_570_, and PRI_norm_ were the best suited for water stress detection in greenhouse tomato production. The results suggest these VIs was not significantly affected by changes in environmental factors.

## 5. Conclusions

This study assessed very high-resolution narrow-band hyperspectral thermal imagery acquired from tomato growth using ASD SPEC 4 in an experiment comprising four water stress treatments (70–100, 60–70, 50–60, 40–50% FC) in sandy loam and silty loam soils for crop water stress assessment. However, the use of reflectance indices for sensing the effects of water stress in tomato plants at different growth stages in sandy loam and silty loam soils revealed that plants grown in each of these soils have different spectral signatures at the different growth stages but displayed similar trends. Quantifying crop leaf water status with remote sensing from the spatial and temporal resolutions it provides imparts highly valued information for making good irrigation management decisions towards optimizing crop productivity. Testing different spectral reflectance indices between 531 to 2130 nm to estimate water stress in tomato growth by comparing various reflectance indices and their relationship with water stress indicators showed that the NDVI, RDVI, WI, NDWI, NDWI_1640_, PRI_570_, and PRI_norm_ were the most sensitive indices for detecting water stress within all the growth stages. This implies that spectra reflectance from 531 to 1640 nm band depth is ideal for estimating water stress in tomato growth in sandy loam and silty loam soils in a greenhouse. This technique is capable of timely and accurately estimating actual crop water status from leaves to scheduling irrigation (date of irrigation, and the specific crop that requires the water). This technique is also unique in determining the onset of water stress in crops grown in sandy loam and silty loam soils. In this study, these VIs have accurately estimated water stress in tomato growth in sandy loam and silty loam soils in a greenhouse even though the relationships between water stress indicators and the selected indexes were high. Further studies on tomato growth in different agroecological zones need to be conducted. This was beyond the scope of this study.

## Figures and Tables

**Figure 1 sensors-21-05705-f001:**
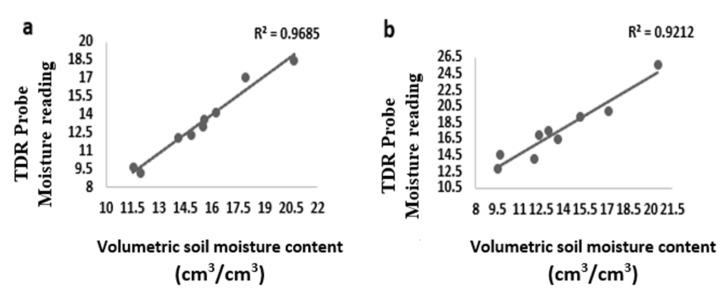
Difference in moisture content determination by volume and TDR (±5%) for (**a**) sandy loam soil and (**b**) silty loam soil.

**Figure 2 sensors-21-05705-f002:**
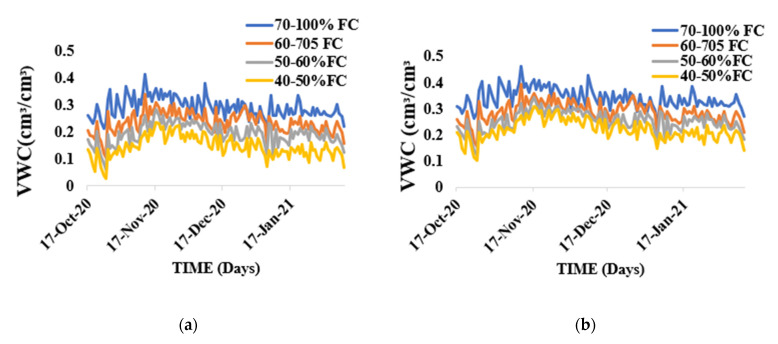
Disparity in soil moisture content during the growing season for sandy loam soil and silty loam soil. (**a**) Sandy loam soil, (**b**) silty loam soil.

**Figure 3 sensors-21-05705-f003:**
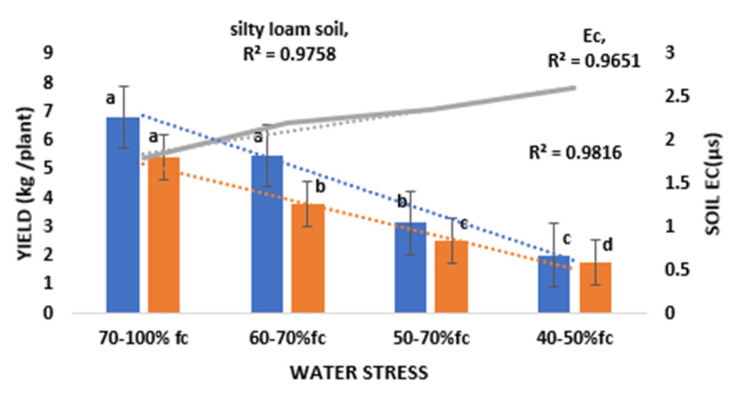
Relationship between sandy loam, silty loam, Ec, and water stress (same alphabet means no significant difference).

**Figure 4 sensors-21-05705-f004:**
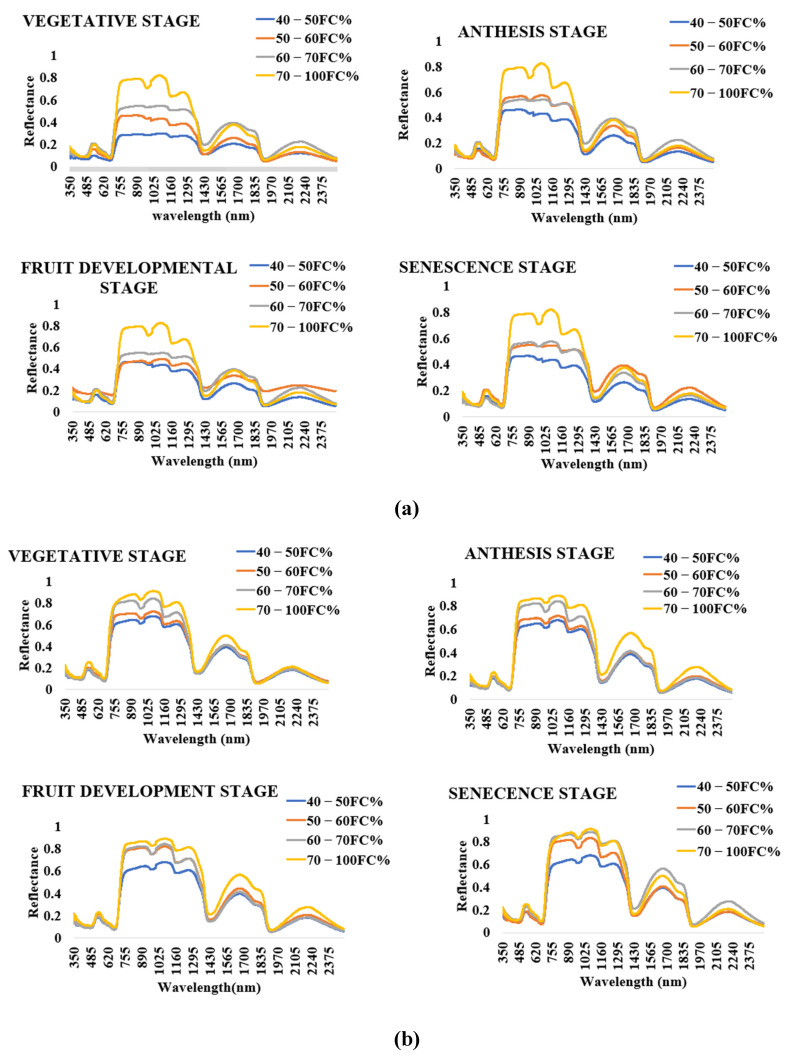
Spectral signatures of the tomato plant leaves at different growth stages for different % FC level for silty loam soil and sandy loam soil: (**a**) spectral signatures of the tomato plant leaves at different growth stages for different % FC for silty loam soil; (**b**) spectral signatures of the tomato plant leave at different growth stages for different % FC level for sandy loam soil.

**Figure 5 sensors-21-05705-f005:**
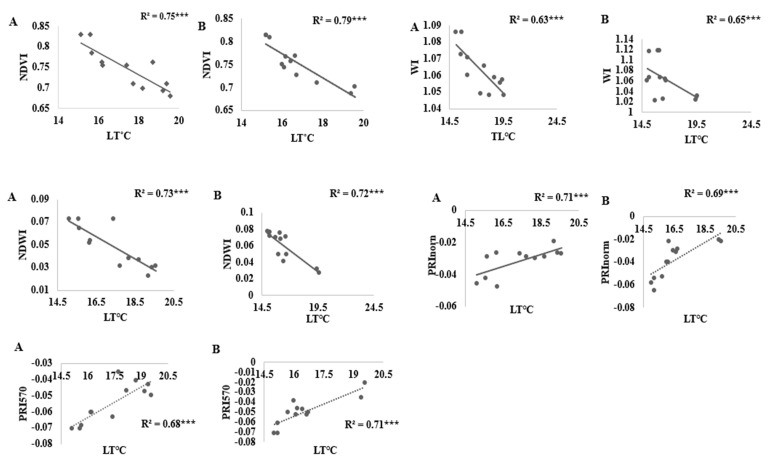
Relationship between LT °C and NDVI, RDVI, WI, NDWI, PRI570, PRInorm, derived from the various treatments in sandy loam and silty loam soils. (A) denotes sandy loam soil, while (B) denotes silty loam soil.) (*** denotes a *p* 0.001 association).

**Table 1 sensors-21-05705-t001:** Soil physicochemical parameters of the experimental site at a depth of (0–30 cm).

Parameters	Sandy Loam Soil	Silt Loam Soil
PH	5.64 ± 0.12	5.30 ± 0.20
O.M (g/kg)	15.91 ± 1.3	22.97 ± 1.79
Total N (g/kg)	1.23 ± 0.013	1.518 ± 0.021
Total P (g/kg)	0.88 ± 0.21	0.865 ± 0.32
Total K (g/kg)	9.30 ± 0.025	19.59 ± 0.05
Alkalized N(mg/kg)	450.28 ± 1.9	72.71 ± 2.38
Avail. P (mg/kg)	195.72 ± 2.43	28.25 ± 1.72
Avail. K (mg/kg)	428.43 ± 42.3	85.50 ± 20.1
Sand (%)	47.6 ± 0.06	43.53 ± 0.34
Clay (%)	17.3 ± 1.54	16.53 ± 1.5
Silt (%)	35.1 ± 0.3	39.93 ± 0.67
Bulk density (gcm^−3^)	1.34 ± 0.11	1.32 ± 0.07
Field capacity (%)	0.21 ± 0.01	31 ± 1.53
WP (%)	0.09 ± 0.13	19 ± 0.19
Saturation point (%)	0.48 ± 0.15	45.73 ± 0.252

**Table 2 sensors-21-05705-t002:** Spectral vegetative indices used in the study, formulae, and respective references.

Index	Formula	References
Greenness vegetation index Normalized difference vegetation index (NDVI)	$NDVI=\frac{\left(R_{800}-R_{670}\right)}{\left(R_{800}+R_{670}\right)}$	[35,36]
Modified normalized difference vegetation index(mNDVI_750_)	$\frac{\left(R_{750}-R_{705}\right)}{\left(R_{750}+R_{705}-\left(2\times R_{445}\right)\right)}$	[6]
Renormalized difference vegetative index (RDVI)	$\frac{\left(R_{800}-R_{670}\right)}{{\left(R_{800}+R_{670}\right)}^{0.5}}$	[28]
Green chlorophyll index (CL*_green_*)	$CL_{green}=\left(\frac{R_{750}}{R_{550}}\right)-1$	[29]
Water content vegetative index Water index (WI)	$\frac{R_{900}}{R_{970}}$	[31]
Simple ratio water index (SRWI)	$\frac{R_{860}}{R_{1240}}$	[17]
Normalized difference water index	$\frac{\left(R_{860}-R_{1240}\right)}{\left(R_{860}+R_{1240}\right)}$	[6]
Normalized difference water index centered at 1640 nm (NDWI_1640_)	$\frac{\left(R_{858}-R_{1640}\right)}{\left(R_{858}+R_{1640}\right)}$	[32]
Normalized difference water index centered at NDWI_2130_	$\frac{\left(R_{858}-R_{2130}\right)}{\left(R_{858}+R_{2130}\right)}$	[32]
Xanthophyll pigment Photochemical reflective index (PRI_570_)	$\frac{\left(R_{570}-R_{531}\right)}{\left(R_{570}+R_{531}\right)}$	[10]
Normalized photochemical reflective index (PRI_norm_)	$\frac{PRI}{\left(RDVI*\left(\frac{R_{700}}{R_{670}}\right)\right)}$	[10]

*R* represents reflectance at a given wavelength (nm).

**Table 3 sensors-21-05705-t003:** Irrigation water applied to tomatoes throughout the growing cycle for each growth stage for the different treatments in sandy loam soil and silty loam soil.

Treatment (% FC)	Duration (Days)		Irrigation Water Applied (mm)							
	Soil A	Soil B	70–100		60–70		50–60		40–50	
			Soil A	Soil B	Soil A	Soil B	Soil A	Soil B	Soil A	Soil B
Vegetative stage	30	32	50	55	32.5	35.8	27.5	30.3	22.5	24.8
Anthesis stage	40	40	60.1	65	39.1	42.3	33.1	23.3	27.1	29.3
Fruit expansion stage	50	50	70	73	45.5	47.5	38.5	40.2	31.5	32.9
Senescence stage	30	31	45.8	50	29.8	32.5	25.2	27.5	20.6	22.5
Total	150	153	225.9	243	146.9	158.1	124.3	121.3	101.7	109.5

**Table 4 sensors-21-05705-t004:** Coefficient of determination (*R*^2^) of the linear relationship RWC (%), LAI (m), LT (°C) LCC, and yield (kg plant^−1^), and vegetation indices computed from the hyperspectral sensor.

Vegetative Indices	Water Stress Indicators in Sandy Loam Soil					Water Stress Indicators Silty Loam Soil				
	LT (℃)	RLWC (%)	LAI (m^2^/m^2)^	LCC (mg/g)	Yield (kg/plant)	LT (℃)	RLWC (%)	LAI (m^2^/m^2^)	LCC (mg/g)	Yield (kg/plant)
Greenness vegetation index NDVI	0.75 ***	0.59 ***	0.80 ***	0.88 ***	0.72 ***	0.79 ***	0.67 ***	0.70 ***	0.87 ***	0.74 ***
RDVI	0.61 ***	0.52 ***	0.63 ***	0.71 ***	0.80 ***	0.68 ***	0.77 ***	0.66 ***	0.65 ***	0.68 ***
CL_green_	0.57 ***	0.45 **	0.47 **	0.62 ***	0.61 ***	0.59 ***	0.36 *	0.22 *	0.60 ***	0.71 ***
Water content vegetative index WI	0.63 ***	0.64 ***	0.53 ***	0.41 **	0.51 **	0.64 ***	0.67 ***	0.57 ***	0.52 ***	0.57 ***
NDWI	0.73 ***	0.72 ***	0.52 ***	0.83 ***	0.74 ***	0.72 ***	0.75 ***	0.55 ***	0.87 ***	0.79 ***
NDWI_1640_	0.64 ***	0.61 ***	0.58 ***	0.75 ***	0.67 ***	0.67 ***	0.54 ***	0.52 ***	0.7 ***	0.71 ***
NDWI_2130_	0.51 **	0.37 *	0.57 **	0.21 *	0.49 **	0.67 ***	0.46 *	0.37 *	0.44 *	0.51 **
Xanthophyll pigment PRI_570_	0.68 ***	0.74 ***	0.45 *	0.55 ***	0.70 ***	0.71 ***	0.70 ***	0.37 *	0.62 ***	0.66 ***
PRI_norm_	0.71 ***	0.73 ***	0.58 **	0.45 *	0.70 ***	0.69 ***	0.69 ***	0.54 **	0.54 **	0.61 ***

* *p* < 0.05; ** *p* < 0.01; *** *p* < 0.001.

## Data Availability

Not applicable.

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
