# Peer review of "Ground-Based Hyperspectral Remote Sensing for Estimating Water Stress in Tomato Growth in Sandy Loam and Silty Loam Soils"

_sensors, 2021, doi:10.3390/s21175705_

Round 1
Reviewer 1 Report
Review for “Ground-based hyperspectral remote sensing for estimating water stress in tomato growth in sandy loam and silty loam soils”.
The study assumes an effective methodology and experimental design to evaluate the relationship among indices derived from hyperspectral spectrometer for estimating water stress in tomato growth in two soil types exposed to four water stress levels.
My recommendation is to accept after minor revision.
General comments:
Firstly, I agree with the authors about the use of various spectral indices. Some indices are most applied throughout the years, due to simple formulation, spectral bands widely common to most launched sensors, and composite products. Most of them are sold as a crop management product for farmers that decide to adhere to precision agriculture or technology in their croplands. The application in these cases (out-of-academy precision agriculture or commercialized precision agriculture) is much more based on default and established preconditions than local or regional tests. Based on diverse papers, we can note that it also largely occurs in the academy.
Likewise, the knowledge that green-based indices outperforms others to detect crop chlorophyll content and water stress in a wide range of agricultural contexts is not a novelty. Nowadays, researchers can explore more than NIR and RED relationships to attest crop chlorophyll content and other vegetational variables with more advanced technologies. Some early studies can add value to your discussions:
Gitelson, A. A., Kaufman, Y. J., & Merzlyak, M. N. (1996). Use of a green channel in remote sensing of global vegetation from EOS-MODIS. Remote Sensing of Environment, 58(3), 289–298.
Marino, S., & Alvino, A. (2015). Hyperspectral vegetation indices for predicting onion (Allium cepa L.) yield spatial variability. Computers and Electronics in Agriculture, 116, 109–117.
Kerkech, M., Hafiane, A., & Canals, R. (2018). Deep leaning approach with colorimetric spaces and vegetation indices for vine diseases detection in UAV images. Computers and electronics in agriculture, 155, 237-243.
Chaves, M., Picoli, M., & Sanches, I. (2020). Recent Applications of Landsat 8/OLI and Sentinel-2/MSI for Land Use and Land Cover Mapping: A Systematic Review. Remote Sensing, 12(18), 3062.
Khanal, S.; KC, K.; Fulton, J.P.; Shearer, S.; Ozkan, E. Remote Sensing in Agriculture—Accomplishments, Limitations, and Opportunities. Remote Sensing. 2020, 12, 3783.
Sishodia, R.P.; Ray, R.L.; Singh, S.K. Applications of Remote Sensing in Precision Agriculture: A Review. Remote Sens. 2020, 12, 3136.
Introduction:
- Formulated and structured in a unique paragraph. Needs to be corrected.
Material and methods:
- A flowchart is recommendable.
- About the indices: it is recommended to confirm each authorship.
- The study area seems small. The authors are concluding the higher suitability of some VIs above others based on a small experiment. It is not necessarily an error; however, this situation should be clear in the text. What is the relevance of this size’s farm for this region? This is the characteristic of the region? Most of the country’s farms are of this size? The authors should explain the reasons to select this study area and discuss if the adopted approach could be applied to other study areas, especially large ones.
- Also, the authors should describe the edaphoclimatic conditions of the region and the tomato crop calendar. A Figure showing the conditions of the study area is recommendable.
- Are crop management or edaphoclimatic conditions significantly varying? There are control areas with fertilization or no fertilization? Edaphoclimatic and crop management practices may be important conditioning factors.
- The experimental design deserves a figure to improve understanding.
- Why only tomatoes? Is it the most planted crop in the region?
- There is a lack of a clear explanation of the method (how is defined and computed). From what is written, it is not possible to understand how it works. As an example: “Soil water content was measured using the TDR and the Gravimetric technique.” Based on a specific methodology? It can be cited. The same for the fertilizer application.
- Other statistical analysis different from Pearson, ANOVA or Fisher was assessed?
- A recommendation is to emphasize the novelty of the current work, making a clear distinction of the assumptions to highlight originality.
Results and Discussion:
- The study seems to be improved by joining Results and Discussion in a unique section. This unique section would be more representative. I strongly suggest the union of these sections.
- As an example, the Authors can discuss confounding factors such as solar/view zenith angle and LAI that also affect canopy reflectance. I think that, if included, its discussion can add reliability to your study.
- Here, a remote sensing-based discussion considering the spectral bands’ characteristics and the tomato can clarify the importance of green band and green-based VIs. It is important to add value to your results and to contextualize your experiment and assumptions.
Conclusion:
- You should explain better why those indices performed better before highlights them in your Conclusions section.
Reviewer 2 Report
Abstract: too long, it has to be briefed e.g., from line 24 to line 42 has to brief and rewritten.
Introduction; also need to be shorter
Material and methods
Line 168; the floor covered with black woven polyethylene mulch. Why black? it may affect the temperature.
Results
Line 376: Fig 3 shows that soil moisture content in sandy loam soil more than in silty loam soil. Are you sure? Normally silty loam has more moisture content than sandy loam soil.
Author Response
Please see the attachment

This manuscript is a resubmission of an earlier submission. The following is a list of the peer review reports and author responses from that submission.